# Incorporating usability evaluation into iterative development of an online platform to support research participation in Parkinson's disease: a mixed methods protocol

Rebecca Chapman ,[1] Marie-Louise Zeissler ,[1] Edward Meinert ,[2,3] Stephen Mullin,[1] Sue Whipps,[4] John Whipps,[4] Kate Hockey,[4] Philip Hockey,[4] Camille B Carroll  [1,3]

¹Applied Parkinson's Research Group, Faculty of Health, University of Plymouth, Plymouth, UK
²Centre for Health Technology, Faculty of Health, University of Plymouth, Plymouth, UK
³Translational and Clinical Research Institute, Newcastle University, Newcastle upon Tyne, UK
⁴Patient and Public Involvement (PPI) Representative, University of Plymouth, Plymouth, UK

**Correspondence to**
Dr Camille B Carroll;
camille.carroll@newcastle.ac.uk

## ABSTRACT

**Introduction** Many people with Parkinson's (PwP) are not given the opportunity or do not have adequate access to participate in clinical research. To address this, we have codeveloped with users an online platform that connects PwP to clinical studies in their local area. It enables site staff to communicate with potential participants and aims to increase the participation of the Parkinson's community in research. This protocol outlines the mixed methods study protocol for the usability testing of the platform.

**Methods and analysis** We will seek user input to finalise the platform's design, which will then be deployed in a limited launch for beta testing. The beta version will be used as a recruitment tool for up to three studies with multiple UK sites. Usability data will be collected from the three intended user groups: PwP, care partners acting on their behalf and site study coordinators. Usability questionnaires and website analytics will be used to capture user experience quantitatively, and a purposive sample of users will be invited to provide further feedback via semistructured interviews. Quantitative data will be analysed using descriptive statistics, and a thematic analysis undertaken for interview data. Data from this study will inform future platform iterations.

**Ethics and dissemination** Ethical approval was obtained from the University of Plymouth (3291; 3 May 2022). We will share our findings via a 'Latest News' section within the platform, presentations, conference meetings and national PwP networks.

## STRENGTHS AND LIMITATIONS OF THIS STUDY

⇒ A mixed methods approach will use both qualitative and quantitative methods and enhance our understanding of any usability issues identified in the development of the platform.
⇒ We will seek feedback on usability from recruiting staff at study sites as well as patients.
⇒ Purposive sampling for semistructured interviews will ensure inclusivity in terms of demographics, geographical location and digital literacy, to ensure issues are identified from a broad range of users.
⇒ Platform users may not have used the whole website prior to the interview, and so their answers may not capture the entirety of the platform.

## INTRODUCTION

Delays in reaching recruitment targets represent a major challenge for clinical trials.[1] A reduction in in-person clinic attendance with the introduction of new, remote care delivery models following the COVID-19 pandemic has further exacerbated this problem.[2] Parkinson's disease (PD) trials generally do not recruit representative populations, and therefore their results are not generalisable, which risks perpetuating healthcare inequalities.[3] Strategies to improve recruitment to trials have been evaluated, but other than telephone reminders and opt-out strategies, very few have been found to be effective.[4] The creation of an online recruitment tool to facilitate communication with and recruitment of research-interested PD patients has the potential to increase the efficiency of recruitment to PD studies.

In 2007, we developed a paper-based register of research-interested PD patients within the South West of England. The Parkinson's Register of the Dementias and Neurodegenerative Diseases Research Network (PRO-DeNDRoN) has been previously evaluated and recognised as a successful and resource efficient recruitment tool, with 85% of registered PRO-DeNDRoN recruiters reporting that the register was a useful means of facilitating research and providing data for planning of service provision.[5] However, this required manual data entry resulting in administrative burden and highlighted the

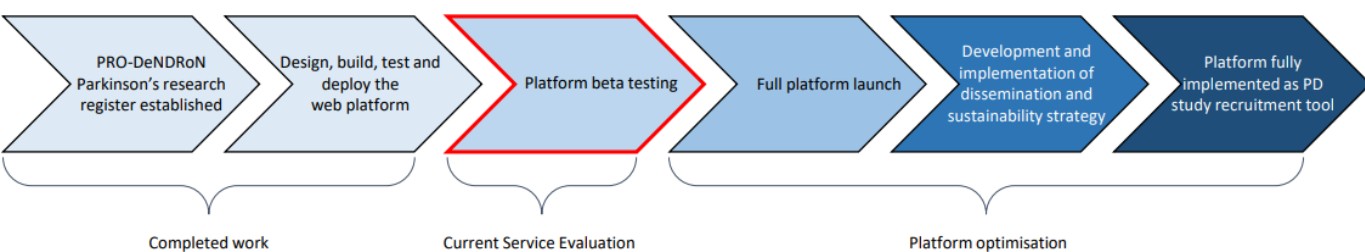

**Figure 1** Project milestones. PD, Parkinson's disease; PRO-DeNDRoN, Parkinson's Register of the Dementias and Neurodegenerative Diseases Research Network.

need for a more efficient, online recruitment platform that is easily accessible to people with Parkinson's (PwP) and time efficient for trial delivery staff.

Online platforms have been used in other disease areas as a successful means of trial recruitment. For example, 'Join Dementia Research' (JDR) is a website that connects people with dementia to research projects across the UK.[6] The website was established in 2015 and in December 2021, JDR reported to have a population of nearly 50 000 research-interested people registered, 12% of whom had a self-declared diagnosis of dementia, with 51% of these being women.[7] This highlights the potential of online registers as a useful tool for disease-specific recruitment.

Originating from the PRO-DeNDRoN register, we have developed an online platform, conceptualised and designed with PwP and care partner input (see figure 1 for project milestone overview). Our platform aims to connect PwP with research projects in their preferred geographical locations. The platform matches participants' eligibility and preferences to open studies and also enables site coordinators to communicate directly with potential participants. The platform has been designed with multi-account permissions in place: central study coordinators (known as 'researchers') who can request to upload a study, administrative staff who review the study documentation, PwP wishing to engage with research and care partners, who are able to set up an account in order to register PwP for studies on their behalf, if the PwP does not wish to, or is not able to, use the platform themselves; care partners are not able to register for studies themselves. Care partners in this context are unpaid and are defined as 'the primary person who feels responsible for, and supports, the PwP'. There is also an account for study coordinators at each individual study site (created automatically once a study has been approved), who can engage with interested

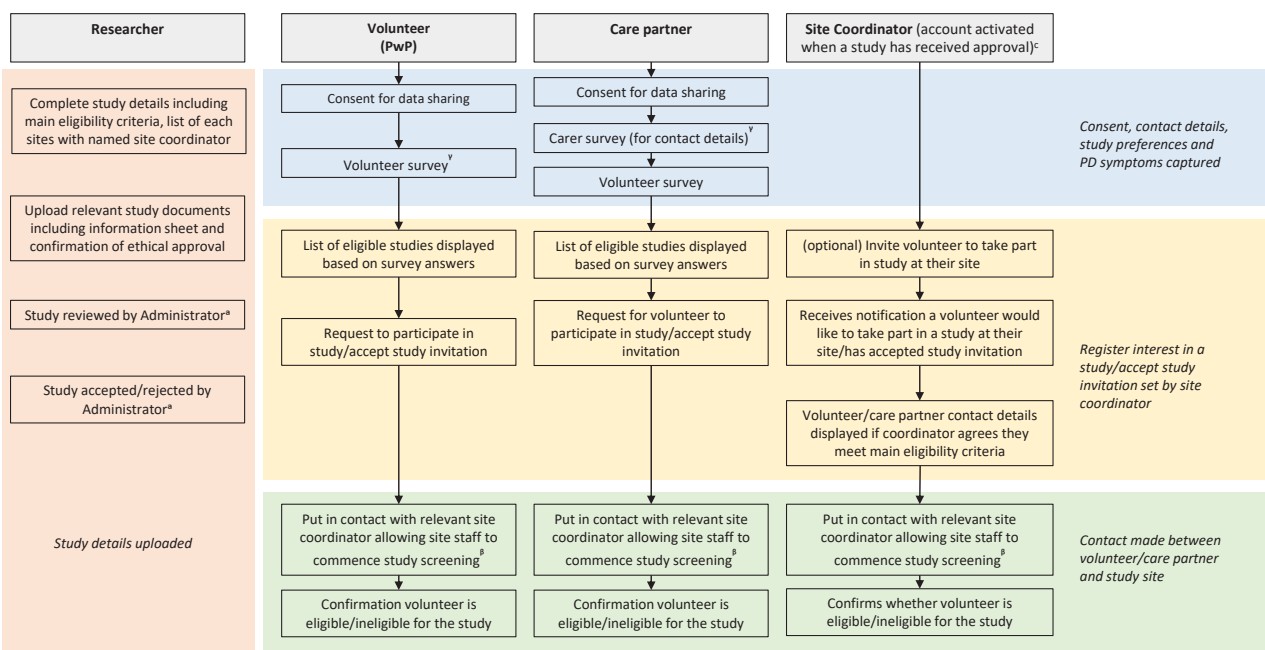

**Figure 2** Website Functionality. [a]Tasks completed by the project Administrator on a separate account. [β]Study screening is performed externally to the website. [γ]Timepoint website user is invited to take part in an interview. PD, Parkinson's disease; PwP, people with Parkinson's.

or eligible participants (see figure 2 for an overview of website functionality).

This project aims to evaluate platform usability and accessibility in the three main user groups (PwP, care partners and study site coordinators), to understand barriers and facilitators to engagement and use.

## METHODS
### Project design

We will invite usability feedback from PwP, care partner and site coordinator representatives to identify any major bugs or functionality issues in the platform prior to beta testing. The platform will then be deployed in a limited launch to enable mixed-methods evaluation via validated usability questionnaire, website analytics and semistructured interviews.

### Dissemination of platform beta-version

Prior to deployment, we will select up to three multicentre studies in PD actively recruiting within the UK, which have good geographical spread of sites. We will invite research investigator teams to input their study details onto the platform. The beta-version of the platform will then be disseminated to the Parkinson's community via national patient and carer networks, as well as charity stakeholders, such as Parkinson's UK and Cure Parkinson's, which will include steering group meetings and advertisements through research information emails.

### Study recruitment and participant selection
#### Usability questionnaires

All platform users will be invited to complete an online questionnaire, which will be available to complete at any time. This will be visible as a tab on the user account menu (for the PwP and the care partner accounts) or an automated email following account activation (for site coordinators) where users can sign up separately for the questionnaire and semistructured interviews. A link will divert users to Joint Information Systems Committee (JISC) online surveys, displaying an information sheet and e-consent form. A survey will then capture user type, ethnicity, age and socioeconomic status, followed by the usability questionnaire.

#### Website analytics

On first visiting the platform, a pop-up will be displayed to ask if the user consents to the use of analytical website cookies. If the user consents, measurement of key performance indicators (KPIs) will be captured.

#### Semistructured interviews

All users will be invited to register interest in participating in remotely conducted interviews to gain an understanding of how they experienced aspects of their user journey, as well as whether and how user experience could be improved. A separate question will be displayed on the same tab as for the usability questionnaires or automated email. For each group, the question will ask if they are interested in providing feedback via an interview and contain two links: one to the information sheet and one to the e-consent form. The information sheet will detail the purpose and nature of the interviews, information on what they involve, details regarding free choice and right to withdraw and will describe the retention of data provisions. The information sheet will be accessed via a separate link to the consent form to allow users adequate time to consider the nature of the study and ask any questions they have. Once e-consent has been obtained, a JISC demographic survey will be displayed to facilitate purposive sampling (please see online supplemental files 1 and 2 for a list of demographic questions).

Participants will be interviewed individually and purposively selected from the three intended platform user groups: PwP, care partners and researchers/site study coordinators. PwP and care partners will be selected based on ethnicity, socioeconomic status, disease duration (if applicable) and digital literacy. Researchers and site coordinators will be selected based on ethnicity, site type, PD research experience and digital literacy. If selected for interview, the participant will be contacted by a member of the project team via their preferred method indicated on the registration survey and arrange a suitable time to conduct the interview. If not selected, the user will receive an email explaining this and thanking them for their time.

### Inclusion criteria for all data collection methods
#### PwP/care partners

► Age ≥18 years.
► Diagnosis of PD/care partner of someone with PD (care partners in this context are unpaid and defined as 'the primary person who feels responsible for, and supports, the PwP').
► Prior experience of using a computing device (including but not limited to PC, Mac and Tablet).
► Has access to a desktop or laptop computer with internet connection.
► Willing and able to give informed consent for participation in the project.
► Willing and able to comply with project requirements.

#### Researchers/site study coordinators

► Age ≥18 years.
► Registered as a named central study coordinator and/or site study coordinator for a study registered on the platform.
► Prior computer experience.
► Willing and able to give informed consent for participation in the project.
► Willing and able to comply with project requirements.

The only specified exclusion criteria for all website users are being unable or unwilling to provide informed consent or comply with project requirements.

## Data collection

### Demographic information

Demographic data will be collected for all users who agree to complete the usability questionnaire and/or semistructured interviews. The measurement tools selected will allow for purposive sampling and aim to maximise inclusivity of interview participants. Postcode will be collected to allow for capture of both rural and urban participants, as well as calculation of socioeconomic status via the index of multiple deprivation calculator.[8] This tool provides the official measures of relative deprivation for small areas in England, and the equivalent tools will be used for the devolved nations.[9–11] Digital literacy will be captured by reproducing the Lloyds Bank Basic Digital Skills Measure (2018)).[12] This is a list of 11 digital tasks split over five skills categories. Respondents are classified as having full basic digital literacy skills if they can complete at least one task in each category.

### Usability questionnaires

Questionnaire feedback will be captured using the Tele-health Usability Questionnaire (TUQ),[13] which is suitable for the collection of opinions from both platform user groups (patients and clinical study site staff), and has previously been used in PD patients.[14] It evaluates the usability of telehealth services and is based on six criteria including usefulness, ease of use and learnability, interface quality, interaction quality, reliability and satisfaction and future use. All questions are optional which allows the measure to be tailored to meet the requirements of specific digital services.

### Semistructured interviews

Semistructured qualitative interviews will be conducted by a University of Plymouth researcher and take place either over the phone or via teleconferencing software and will last no longer than 1 hour. The participant will be facilitated through the interview process, and prompts will be used to enable guided conversations using an interview guide (see online supplemental files 3 and 4 for prototype interview questions). This guide has been informed by the mHealth for older users (MOLD-US) framework,[15] which identifies four key categories of ageing barriers which influence the usability of health technologies, and that are of particular relevance in PD: cognition, physical ability, perception and motivation. This allows for results to be classified and interpreted based on impediments that are intrinsic to usability issues experienced by older adults. The topic guide will also be informed by the initial usability feedback prior to live deployment.

The guide contains questions that seek feedback on each aspect of website functionality, from user registration to confirmation of study eligibility (if applicable) and participants will be given the opportunity to identify what went well or what could be improved. All interview sessions will be recorded via video and/or audio capture and will be fully transcribed by a University of Plymouth researcher.

### Website analytics

KPIs of user behaviour will be measured using HotJar analytical software,[16] where users consent for this to be captured through the acceptance of analytical website cookies. General user behaviour will be captured through the use of heatmaps, and individual user journeys will be mapped and navigation timings recorded.

Further performance indicators will include the number of PwP registered to the platform, the percentage requesting and/or invited to take part in a study and the percentage of PwP accepted for study screening and enrolment.

## Sample size

### Telehealth Usability Questionnaire (TUQ)

We determined a 95% CI, an accuracy of ±5 percentage units and a satisfaction with the website random estimation of 50%. Taking into account a target population limit of 500 participants that can be supported within the test server the sample size needed is 218 participants.[17]

### Web analytics

We will gather data from all users who consent to the use of website analytics, but we will aim for 10 per user group, per data capture method.

### Semistructured interviews

A purposive sample of 20 users will be selected for semistructured interviews (10 per participant group). While still allowing for a richly textured understanding of the usability issues,[18] it is maintained that little new information is generated after completing 20 qualitative interviews.

## Data analysis

### Quantitative data

Median (range) data will be collated on all usability questionnaire items as an indicator of usability levels for both user groups. For website analytics, average timings to complete user journey subtasks and individual recorded user journeys of 'slow' and 'fast' completers will be descriptively analysed to gain insights into processes that may cause difficulties. Feedback-specific analysis will also be undertaken and, depending on interview feedback, particular parts of website user journeys may be descriptively analysed further to gain insights into how optimal improvements can be made.

### Qualitative data

For initial feedback prior to beta-testing, we will evaluate the risk of harm according to standard processes,[19] which comprise an evaluation of task criticality, frequency and impact. Ratings with high severity of harm ratings will be prioritised for amendment over issues with less high ratings.

Interview data will be stored, managed and analysed with NVivo. Thematic analysis will be undertaken using the six-step approach of Braun and Clarke: (1) become familiar with data, (2) generate initial codes, (3) search

for themes, (4) review themes, (5) define themes and (6) write-up.[20]

## Project management

A project management team consisting of researchers, clinicians and patient and public involvement representatives will meet monthly and take on the role of data monitoring and project conduct.

All project data will be managed in line with local and national General Data Protection Regulation (GDPR) requirements. All digital data (including digital consent forms) will be stored on University of Plymouth OneDrive as access requires a university username and password. Backups will be made on a University staff computer hard disk drive as these are located on University laptops that require a username a password. All data will only be accessed by project staff and will be anonymised and only be identified by a study ID number. Name and study ID numbers will be stored securely on a separate tracking sheet. All data will be archived for 10 years following study completion. On completion of the 10-year archive period, and following confirmation from the sponsor and Chief Investigator (CI), all digital data will be destroyed.

## Patient and public involvement

PwP and their care partners have been represented in the project group and have been since the project's inception. They have contributed to website and study design and will be providing feedback to finalise the design of the platform prior to beta-testing. They have been and will continue to be responsible for contributing to all patient and public facing materials relating to the project and the dissemination of its findings.

## Ethics and dissemination

The University of Plymouth Faculty of Health Research Ethics and Integrity Committee (Ref. 3291) approved the use of interviews to capture usability feedback on 3 May 2022. The university will also act as project sponsor.

Any protocol modifications will be reported on the website. The project team will prepare a plain English summary of the usability evaluation results, which will also be displayed on the website and sent to the users who took part in beta-testing. The final results of the project will be disseminated via presentations at appropriate scientific meetings and conferences and publication in appropriate peer-reviewed journals, as well as dissemination within the Parkinson's patient community.

## DISCUSSION

The aim of the PD research platform is to increase the communication and participation of PwP in health and care research and to reduce the administrative burden involved in enabling this participation. There is an urgent need to address the challenge of recruiting PwP to research studies. Web-based platforms can increase the efficiency of recruitment to PD studies, helping to ensure that recruitment targets are met within planned time-frames. This usability project undertakes robust usability evaluation of a new online research matching platform, something that has not previously been created specifically for PwP.

Inclusivity of participants is a particular issue in PD studies, particularly in terms of age, social deprivation, gender and ethnicity,[21] which has a major effect on the generalisability of trial findings. The National Institute for Health and Care Research's improving inclusion of underserved groups in clinical research (NIHR-INCLUDE) project highlights the multidimensional and intersectional nature of the inclusion of under-served groups and defines examples of potential barriers, such as a lack of available trials and poor trial promotion.[22] We expect our web-based tool to support inclusivity; we will therefore ensure that our evaluation covers the breadth of the workforce and potential participants with regard to geographical location, demographics, ethnicity and socioeconomic factors. Digital literacy will also be captured as this been identified as a potential challenge in the use of digital technologies within healthcare.[23] Furthermore, it is an important factor to capture in online usability assessments, particularly those involving older adults, as level of education or digital literacy are likely to influence how the user perceives the usability of the platform.[24]

Although digital healthcare tools have been developed in conjunction with PwP,[25] as well as those with other neurodegenerative diseases such as Alzheimer's,[26] developing digital solutions for PwP, and in particular those who are older, still presents specific challenges related to both age and disease, and so these need to be considered so that the platform matches the users' needs and characteristics. By using the MOLD-US framework to inform the interview topic guide, it allows these challenges to be addressed. The framework has been previously used to assess usability barriers in older adults in the evaluation of health technologies in other disease areas and to allow for broader representation of the general ageing population[26] and identifies the following key categories of ageing barriers which influence the usability of health technologies.

1. Cognition—as one of the most common non-motor features of PD,[25] cognitive impairment can influence memory, processing speed and attention.[27]
2. Physical ability—PD is characterised by motor impairments such as bradykinesia, muscular rigidity and tremor.[28] Slower movements and tremor may impact the speed of performance and increase error rate, leading to less subjective satisfaction.[26]
3. Perception—visual impairments in PD include factors such as colour perception; the visual effectiveness of certain colours can therefore compromise usability performance.[29]
4. Motivation—up to 70% of PwP experience apathy,[30] resulting in reduced interest and initiative, If the perceived value and ease of use of a technology interface

is not immediate, then older adults are much less likely to use it in the future.[31]

There have been misconceptions reported by research teams for other online platforms such as JDR, including location of website users, a lack of awareness in contacting potential participants and the context of where the platform sits in the wider National Health Service (NHS) recruitment landscape.[32] Therefore, to ensure all users and stakeholders have full understanding of the platform's performance and functionality, as well as enabling platform optimisation as a clinical Parkinson's research trial recruitment tool in the future, the capture of KPI such as national uptake, inclusivity and recruitment performance over time will be crucial. This evaluation has important implications for the availability of research opportunities for PwP, and our mixed methods approach will help to enhance the understanding of any usability issues identified in the development of the platform. Seeking feedback from recruiting staff at study sites as well as PwP and care partners will help maximise the functionality and accessibility of an online platform that is tailored to the needs of both patients and study staff. Purposive sampling for semistructured interviews will ensure inclusivity in terms of demographics, geographical location and digital literacy. However, by using a remote asynchronous method for evaluating usability, we are not able to confirm whether each platform account is being used by a singular user and, in turn, their TUQ responses. Furthermore, platform users may not have used the whole website prior to the interview, and so their answers may not capture the entire platform functionality.

Our platform has the potential to increase the efficiency of recruitment to PD studies, helping to ensure that recruitment targets are met for interventional trials within their planned timeframes. As a UK-wide platform, it will also support inclusivity in trial recruitment, thereby facilitating more representative and generalisable trial results.

**Correction notice** This article has been corrected since it was published. Licence changed to CC BY on 15/01/24.

**Acknowledgements** The authors would like to thank the additional patient and care partner volunteers who gave input to the design of the usability methodology, and Dr Lexy Sorrell who assisted with sample size calculation.

**Contributors** CBC, M-LZ, EM, SM and RC conceived the usability project design, and SW, JW, KH and PH had further input into the design of the usability methodology. RC wrote all the drafts of the protocol with significant input from CBC and M-LZ. EM, SM, SW, JW, KH and PH reviewed and revised the manuscript. All authors approved the final manuscript as submitted.

**Funding** This work was supported by unrestricted grants from Roche, Pfizer, Lundbeck and the Medical Research Council (award grant numbers n/a). The funders did not contribute to the protocol in any way.

**Competing interests** None declared.

**Patient and public involvement** Patients and/or the public were involved in the design, or conduct, or reporting, or dissemination plans of this research. Refer to the Methods section for further details.

**Patient consent for publication** Not applicable.

**Provenance and peer review** Not commissioned; externally peer reviewed.

**ORCID iDs**
Rebecca Chapman http://orcid.org/0000-0001-7734-6905
Marie-Louise Zeissler http://orcid.org/0000-0002-6232-4284
Edward Meinert http://orcid.org/0000-0003-2484-3347
Camille B Carroll http://orcid.org/0000-0001-7472-953X

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
