## [Reviewer comments · BMJ Open]

ARTICLE DETAILS

TITLE (PROVISIONAL)	Incorporating usability evaluation into iterative development of an online platform to support research participation in Parkinson's disease: A mixed methods protocol
AUTHORS	Chapman, Rebecca; Zeissler, Marie-Louise; Meinert, Edward; Mullin, Stephen; Whipps, Sue; Whipps, John; Hockey, Kate; Hockey, Philip; Carroll, Camille

VERSION 1 – REVIEW

REVIEWER	Wang, You Chongqing Medical University Affiliated Second Hospital
REVIEW RETURNED	22-Aug-2023

GENERAL COMMENTS	It is my great honor to review this manuscript. The authors developed an online platform that connects patients with Parkinson to clinical studies to reduce the administrative burden involved in enabling this participation. It was an interesting work and might impel the development of PD studies. I noticed that in this questionnaire, the questions about Parkinson's symptoms were mostly subjective, it would be more accurate if these questions were more objective. Moreover, since the majority of PWP's are elderly populations, too many subjective questions may make it more difficult for them to answer the questions and ultimately lose these key target patients.
--

REVIEWER	Gerritzen, Esther University of Nottingham, Mental Health and Clinical Neuroscience
REVIEW RETURNED	23-Aug-2023

GENERAL COMMENTS	General comments: This protocol describes a very important piece of work. The impact of this work can go beyond the Parkinson's population only and serve as an example for other conditions that do not have such a platform yet. Some parts require a more explanation, for example about the role of the care partner. The discussion could benefit from a deeper reflection on the challenges and limitations of this project. Below I listed my comments for each section. Title and abstract: 1. For clarity it would be good to mention in the title that this is a study protocol. E.g. 'Incorporating usability evaluation into iterative development of an online platform to support research participation in Parkinson's disease: a study protocol'.2. Strengths and limitations are mentioned in the abstract but these are not addressed in the main manuscript.
--

	Introduction 3. It does not become entirely clear whether the proposed platform is only for people living with a Parkinson's disease diagnosis, or if their care partners can also register themselves on the online platform, e.g., to be linked to studies focusing on carers. It may be good to clarify this. 4. The last sentence of the introduction says '... the two main user groups...'. It would be helpful to mention what the two main user groups are. In the methods section on page 4 the authors describe the user groups as people with Parkinson's disease / their care partners and site study coordinators, however, it would be helpful if this is made clear already here in the introduction. Methods 5. Project design: a. See comment 3 about the main user groups. It is a bit confusing as here it seems that the authors mention three groups (people with Parkinson's disease, care partners, and study site coordinators). b. See comment 2 about the role of care partners. The authors mention that care partners are invited to provide usability feedback. Does this mean that they will also use the proposed platform? Or will they give feedback on behalf of the person with Parkinson's disease? The authors should provide a clear description of why care partners are invited to provide feedback on usability. 6. Dissemination of platform beta-version: It would be good if the authors can already provide some examples of national patient and carer networks and charity stakeholders they will work with. 7. Study recruitment and participant selection: a. 'Usability questionnaires': The authors speak of the user account menu, which is for people with Parkinson's disease and care partners. This relates to my earlier comments on the involvement of care partners. Is this referring to a shared account for people with Parkinson's disease and their care partner? Or can they both have their own personal account? If it is a shared account the authors should explain why, as this relates to the autonomy and independence of the person with Parkinson's disease. A shared account can also suggest that every person with Parkinson's disease has a care partner, which is not always the case. b. 'Semi-structured interviews': The authors state 'PwP / care partners will be selected based on ...'. The authors should make clear whether it will be joint interviews or individual interviews. If they are joint interviews, the authors should state why they decided for this option (see comment 6). The authors should explain (a) how they will ensure the personhood, autonomy and independence of the person with Parkinson's disease, (b) how they will ensure that the voice of the person with Parkinson's disease is heard and maintain a balance between the person with Parkinson's disease and the care partner (as there is a risk that the care partner will speak for the person with Parkinson's disease), and (c) whether people who do not have a care partner can take part. 8. Inclusion criteria: 'has access to a desktop or laptop computer'. What about tablets or smartphones? 9. Data collection: It would be helpful to know why the measurement tools and questionnaires were selected and whether they have been used with people with Parkinson's disease before.
--	---

	This was explained for the MOLD-US framework but not for some of the other tools. Discussion 10. In the first sentence the authors mention that the aim is to increase the participation of people with Parkinson's disease in clinical research. In the second sentence they mention 'research projects'. It would be good if the authors can reflect on whether the platform is only for clinical research, or also for non-clinical research projects (e.g. psychosocial interventions). If it is only for clinical research, the authors should explain why. 11. 'Developing digital solutions for older adults with Parkinson's ...'. Saying 'older adults' excludes people with young onset Parkinson's disease. Even though the majority of people with Parkinson's disease is of older age, it is important also acknowledge the younger people and their specific needs. Perhaps the authors can provide a brief reflection on this in the discussion. 12. The description of the MOLD-US framework may fit better in the methods section than in the discussion. 13. A clear reflection on the limitations of this study and the potential challenges is currently missing from the discussion. One limitation is mentioned in the abstract, however, a deeper reflection is needed. The authors could for example reflect on whether they foresee any challenges with recruitment or data collection. The authors mention that it can be challenging to develop digital solutions for older adults and for people with Parkinson's disease. This requires a more detailed discussion, as research shows that people with Parkinson's disease and older adults can use technology and are involved in eHealth interventions. Perhaps the authors can provide a few examples, supported by literature, of which specific challenges they foresee in this, and what they will do to overcome them.
--	---

VERSION 1 – AUTHOR RESPONSE

Reviewer 1

Reviewer Comment	Response	Page and paragraph number (tracked changes document)
I noticed that in this questionnaire, the questions about Parkinson's symptoms were mostly subjective, it would be more accurate if these questions were more objective. Moreover, since the majority of PWP's are elderly populations, too many subjective questions may make it more difficult for them to answer the questions and ultimately lose these key target patients	Thank you for your comments. We believe the subjective questions in the questionnaire pertain to the familiarity with digital technology questions in Supplemental Material 1. This is the Lloyds Bank Basic Digital Skills Measure (2018), which has previously been deployed as a measurement of digital literacy and will not be used as a measure of Parkinson's symptoms. Details of symptoms will be captured in the registration process for the web platform itself (not detailed in this manuscript), and part of this study will be to evaluate the acceptability of this.	n/a

Reviewer 2

Reviewer Comment	Response	Page and paragraph number (tracked changes document)
1. For clarity it would be good to mention in the title that this is a study protocol. E.g. 'Incorporating usability evaluation into iterative development of an online platform to support research participation in Parkinson's disease: a study protocol'	Thank you so much for your kind and positive comments. This has now been updated as per the formatting change request	p1
2. Strengths and limitations are mentioned in the abstract but these are not addressed in the main manuscript	Thank you for your suggestion, these have now been added in the Discussion section	p8, para 6 p9, para 1
3. It does not become entirely clear whether the proposed platform is only for people living with a Parkinson's disease diagnosis, or if their care partners can also register themselves on the online platform, e.g., to be linked to studies focusing on	We understand that this was not made clear, and would like to clarify that care partners are not able to register for studies themselves, but can set up an account on the platform to register a PwP for studies on their behalf. This has now been explained in the manuscript by adding to the sentence explaining care partners in the introduction: Previous wording: "care partners wishing to register PwP for studies on their behalf"	p3, para 3

carers. It may be good to clarify this.	New wording: “care partners, who are able set up an account in order to register PwP for studies on their behalf, if the PwP does not wish to, or is not able to, use the platform themselves; care partners are not able to register for studies themselves.”	
4. The last sentence of the introduction says ‘... the two main user groups...’. It would be helpful to mention what the two main user groups are. In the methods section on page 4 the authors describe the user groups as people with Parkinson’s disease / their care partners and site study coordinators, however, it would be helpful if this is made clear already here in the introduction	Thank you for your suggestion. Reflecting upon your comments, as the usability experience for care partners will be slightly different than that of PwPs due to slightly differing functionality, we have decided to make care partners a completely separate user group. Therefore, the study will now have three user groups, which we have reflected in the supplemental materials and the manuscript, including the last sentence of the introduction: Previous wording: “This project aims to evaluate platform usability and accessibility in the two main user groups” New wording: “This project aims to evaluate platform usability and accessibility in the three main user groups (PwP, care partners and study site coordinators)”	p3, para 4
5a. See comment 3 about the main user groups. It is a bit confusing as here it seems that the authors mention three groups (people with Parkinson’s disease, care partners, and study site coordinators).	We hope that it is now clear from the comment above that there are three user groups.	p3, para 5
5b. See comment 2 about the role of care partners. The authors mention that care partners are invited to	We apologise that this wasn’t clear in the original manuscript. We hope that by now explaining in the introduction that care partners can set up an account to register for studies on behalf of a PwP, and that	p4, para 2

provide usability feedback. Does this mean that they will also use the proposed platform? Or will they give feedback on behalf of the person with Parkinson's disease? The authors should provide a clear description of why care partners are invited to provide feedback on usability.	they now comprise a separate user group, it is clear that they will also use the proposed platform and why they are invited to provide feedback. Wording in this methods section has also been changed: Previous wording: “All platform users will be invited to complete an online questionnaire which will be available to complete at any time. This will be visible as a tab on the user account menu (for PwP and care partners)” New wording: “All platform users will be invited to complete an online questionnaire which will be available to complete at any time. This will be visible as a tab on the user account menu (for the PwP and the care partner accounts)”	
6. Dissemination of platform beta-version: It would be good if the authors can already provide some examples of national patient and carer networks and charity stakeholders they will work with.	Thank you for giving us the opportunity to provide some clarification. This has now been added to the manuscript: Previous wording: “The beta-version of the platform will then be disseminated to the Parkinson's community via national patient and carer networks, as well as charity stakeholders” New wording: “The beta-version of the platform will then be disseminated to the Parkinson's community via national patient and carer networks, as well as charity stakeholders, such as Parkinson's UK and Cure Parkinson's”	p4, para 1
7a. 'Usability questionnaires': The authors speak of the user	Many thanks for raising this important point. By now explaining in the introduction that care partners have an account to register PwP for studies on their behalf, and that is it for instances when the PwP	p3, para 3

account menu, which is for people with Parkinson's disease and care partners. This relates to my earlier comments on the involvement of care partners. Is this referring to a shared account for people with Parkinson's disease and their care partner? Or can they both have their own personal account? If it is a shared account the authors should explain why, as this relates to the autonomy and independence of the person with Parkinson's disease. A shared account can also suggest that every person with Parkinson's disease has a care partner, which is not always the case.	does not wish to, or is not able to use the platform themselves, it is now clear that this is not a shared account.	
7b. 'Semi-structured interviews': The authors state 'PwP / care partners will be selected based on ...'. The authors should make clear whether it will be joint interviews or individual interviews.	We would like to confirm that these will be separate interviews, and this has now been made clear in the manuscript: Previous wording: "Participants will comprise two intended platform user groups: PwP/their care partners and researchers/site study coordinators"	p4, para 5 Supplemental Material 1

	New wording: “Participants will be interviewed individually and purposively selected from the three intended platform user groups: PwP, care partners and researchers/site study coordinators.” A question has also been added to the interview registration form for this group, to ask whether they are a PwP or care partner	
8. Inclusion criteria: ‘has access to a desktop or laptop computer’. What about tablets or smartphones?	Thank you for this suggestion, however the platform has not yet been tested on tablets or smartphones and so we will be limiting its use to laptops and desktop computers.	n/a
9. Data collection: It would be helpful to know why the measurement tools and questionnaires were selected and whether they have been used with people with Parkinson’s disease before. This was explained for the MOLD-US framework but not for some of the other tools.	We would like to clarify that, although not specifically used in people with Parkinson’s disease, the measurement tools for purposive sampling have been selected in order to try and maximise the inclusivity of our feedback participants. The TUQ has been previously been used with people with Parkinson’s disease. This has been updated in the manuscript. Previous wording: “Demographic data will be collected for all users who agree to complete the usability questionnaire and/or semi-structured interviews.” “Questionnaire feedback will be captured using the Telehealth Usability Questionnaire (TUQ) (13), which is suitable for the collection of opinions from both platform user groups (patients and clinical study site staff)” New wording:	p5, para 4&5

	“Demographic data will be collected for all users who agree to complete the usability questionnaire and/or semi-structured interviews. The measurement tools selected will allow for purposive sampling and aim to maximise inclusivity of interview participants” “Questionnaire feedback will be captured using the Telehealth Usability Questionnaire (TUQ) (13), which is suitable for the collection of opinions from both platform user groups (patients and clinical study site staff), and has previously been used in PD patients (14)”	
10. In the first sentence the authors mention that the aim is to increase the participation of people with Parkinson’s disease in clinical research. In the second sentence they mention ‘research projects’. It would be good if the authors can reflect on whether the platform is only for clinical research, or also for non-clinical research projects (e.g. psychosocial interventions). If it is only for clinical research, the authors should explain why.	Thank you for allowing us to clarify this. This platform aims to increase participation of all health and care research projects, and this has now been reflected in the discussion section: Previous wording: “The aim of the Parkinson’s research platform is to increase the communication and participation of PwP in clinical research” New wording: “The aim of the Parkinson’s research platform is to increase the communication and participation of PwP in health and care research”	p7, para 8
11. ‘Developing digital solutions for older adults with Parkinson’s ...’. Saying ‘older adults’ excludes people with young onset Parkinson’s disease. Even though the majority of people with Parkinson’s	Thank you for your suggestion, this has been reflected in the manuscript so that younger people with Parkinson’s are not excluded: Previous wording:	p8, para 2

disease is of older age, it is important also acknowledge the younger people and their specific needs. Perhaps the authors can provide a brief reflection on this in the discussion.	“Developing digital solutions for older adults with Parkinson’s” New wording: “Developing digital solutions for PwP, and in particular those that are older”	
12. The description of the MOLD-US framework may fit better in the methods section than in the discussion.	Thank you for your suggestion. We have left the original wording in the discussion section as it helps to address comment number 14, but have also added some description of the framework to the methods section: Previous wording: “This guide has been informed by the MOLD-US framework (15), which allows for results to be classified and interpreted based on impediments that are intrinsic to usability issues experienced by older adults, and will also be informed by the initial usability feedback prior to live deployment.” New wording: “This guide has been informed by the MOLD-US framework (15), which identifies four key categories of ageing barriers which influence the usability of health technologies, and that are of particular relevance in PD: cognition, physical ability, perception and motivation. This allows for results to be classified and interpreted based on impediments that are intrinsic to usability issues experienced by older adults. The topic guide will also be informed by the initial usability feedback prior to live deployment”	p5, para 6 p6, para 1
13. A clear reflection on the limitations of this study and the potential challenges is currently missing from the discussion. One limitation is mentioned in the abstract, however, a deeper reflection is needed. The authors could for example	Thank you for this suggestion, more discussion on limitations has now been added to the manuscript: Additional wording: “Purposive sampling for semi-structured interviews will ensure inclusivity in terms of demographics, geographical location, and digital literacy. However, by using a remote asynchronous method for evaluating usability we are not able to confirm whether each platform account is being used by a	p9, para 1

reflect on whether they foresee any challenges with recruitment or data collection.	singular user and, in turn, their TUQ responses. Furthermore, platform users may not have utilised the whole website prior to interview, and so their answers may not capture entire platform functionality”	
14. The authors mention that it can be challenging to develop digital solutions for older adults and for people with Parkinson’s disease. This requires a more detailed discussion, as research shows that people with Parkinson’s disease and older adults can use technology and are involved in eHealth interventions. Perhaps the authors can provide a few examples, supported by literature, of which specific challenges they foresee in this, and what they will do to overcome them.	Thank you for your comment. We feel that by keeping the MOLD-US framework description in this section, it helps to explain what the challenges are that we foresee in users utilising the platform, and that these will then be addressed in the interview topic guide. We have added in some wording to explain that there have been previous digital healthcare tools developed with people with Parkinson’s and older adults: Additional wording: “Although digital healthcare tools have been developed in conjunction with PwP (25), as well as those with other neurodegenerative diseases such as Alzheimer’s (26)”	p8, para 2